# Identification of a molecular basis for the juvenile sleep state

Leela Chakravarti Dilley[1], Milan Szuperak[1], Naihua N Gong[1], Charlette E Williams[1], Ricardo Linares Saldana[2], David S Garbe[1], Mubarak Hussain Syed[3], Rajan Jain[2], Matthew S Kayser[1,4,5]*

[1]Department of Psychiatry, Perelman School of Medicine at the University of Pennsylvania, Philadelphia, United States; [2]Department of Medicine, Perelman School of Medicine at the University of Pennsylvania, Philadelphia, United States; [3]Department of Biology, University of New Mexico, Albuquerque, United States; [4]Department of Neuroscience, Perelman School of Medicine at the University of Pennsylvania, Philadelphia, United States; [5]Chronobiology and Sleep Institute, Perelman School of Medicine at the University of Pennsylvania, Philadelphia, United States

**Abstract** Across species, sleep in young animals is critical for normal brain maturation. The molecular determinants of early life sleep remain unknown. Through an RNAi-based screen, we identified a gene, *pdm3*, required for sleep maturation in *Drosophila*. Pdm3, a transcription factor, coordinates an early developmental program that prepares the brain to later execute high levels of juvenile adult sleep. PDM3 controls the wiring of wake-promoting dopaminergic (DA) neurites to a sleep-promoting region, and loss of PDM3 prematurely increases DA inhibition of the sleep center, abolishing the juvenile sleep state. RNA-Seq/ChIP-Seq and a subsequent modifier screen reveal that *pdm3* represses expression of the synaptogenesis gene *Msp300* to establish the appropriate window for DA innervation. These studies define the molecular cues governing sleep behavioral and circuit development, and suggest sleep disorders may be of neurodevelopmental origin.

*For correspondence:
kayser@pennmedicine.upenn.edu

**Competing interests:** The authors declare that no competing interests exist.

## Introduction

Across species, sleep amounts are highest in early life and decrease as animals mature (*Kayser and Biron, 2016*; *Roffwarg et al., 1966*). Increasing evidence suggests early life sleep may represent a distinct behavioral state, uniquely evolved for the needs of a developing nervous system (*Blumberg, 2015*; *Clawson et al., 2016*; *Dilley et al., 2018*; *Frank, 2011*). In humans, childhood sleep disturbances portend later neurocognitive deficits, possibly because sleep loss impinges on neural circuit formation (*Kotagal, 2015*; *O'Brien, 2009*). Although mechanisms controlling mature adult sleep have been uncovered (*Allada et al., 2017*), the regulation of early life sleep remains poorly understood.

Like other animals, the fruit fly, *Drosophila melanogaster*, exhibits increased sleep duration in young adulthood that tapers with maturity (*Dilley et al., 2018*; *Kayser et al., 2014*; *Shaw et al., 2000*). At the circuit level, activity of wake-promoting dopaminergic (DA) neurons increases as flies mature, exerting greater inhibitory influence on the sleep-promoting dorsal fan shaped body (dFSB) (*Donlea et al., 2014*; *Kayser et al., 2014*; *Liu et al., 2012*; *Ueno et al., 2012*). The mechanistic underpinnings of how this central sleep circuit develops are not defined. More broadly, there are no known genes that regulate sleep ontogenetic change. We previously found that all studied short and long-sleeping *Drosophila* mutants sleep more when young (*Dilley et al., 2018*). Thus, genes regulating sleep ontogenetic change are likely distinct from those that control sleep duration.

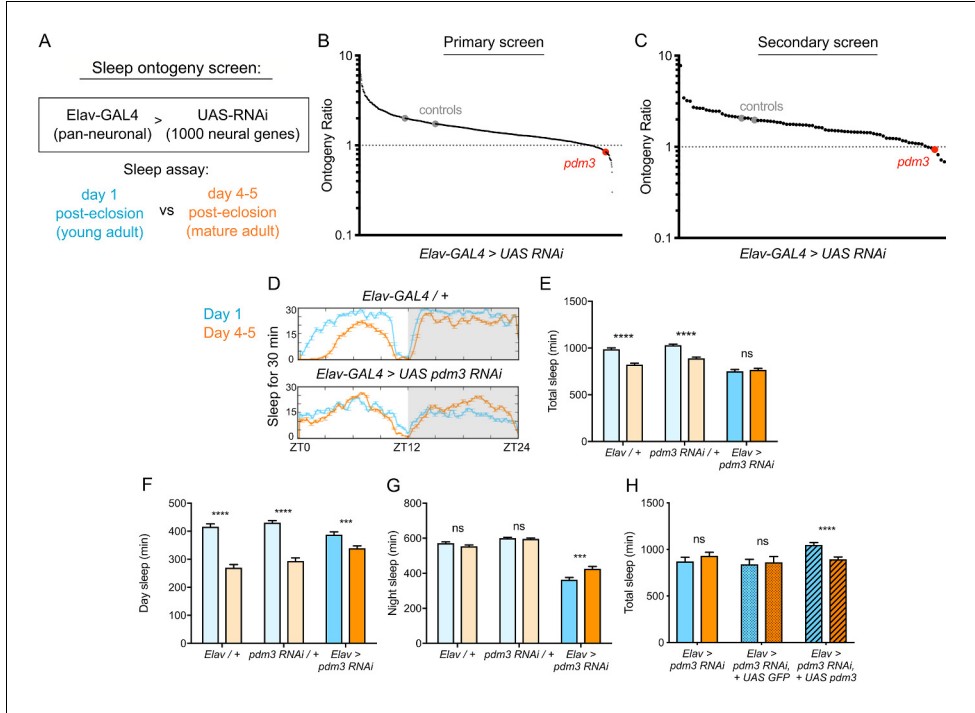

**Figure 1.** *Pdm3* controls sleep ontogeny in *Drosophila*. (**A**) Ontogeny screen design (**B**) Primary sleep ontogeny screen. Ontogeny ratio = (min daytime sleep, young) / (min daytime sleep, mature). (**C**) Secondary screen of primary hits (n ≥ 8 flies per genotype/age in B and C). (**D**) Representative sleep traces of genetic controls (top) and *pdm3* knockdown (bottom). Young flies are shown in blue and mature flies are shown in orange. Comparison of (**E**) total sleep, (**F**) day sleep and (**G**) night sleep in *pdm3* RNAi and controls at day 1 versus day 4–5 (n = 97, 106, 119, 95, 140, 145 left to right in E-G). (**H**) Total sleep time with re-expression of PDM3 (right) versus a control UAS-GFP construct (middle) (n = 24, 24, 19, 16, 30, 31 left to right). Graphs in this figure and all others unless otherwise specified are presented as means ± SEM. ****p<0.0001, ***p<0.001, **p<0.01, *p<0.05; multiple Student's *t* tests with Holm-Sidak correction, alpha = 0.05 (**E–H**).

The online version of this article includes the following figure supplement(s) for figure 1:

**Figure supplement 1.** Young versus mature sleep amounts in primary and secondary sleep ontogeny screens.
**Figure supplement 2.** Confirmation of *pdm3* knockdown.

From an RNAi-based screen, we identified the transcription factor *pdm3* as a genetic regulator of sleep ontogeny. In contrast to all other characterized sleep mutants, flies lacking PDM3 do not attain appropriately high juvenile sleep amounts. *Pdm3* knockdown specifically during pupal development prematurely increases wake-promoting DA input to the dFSB. Blocking DA signaling to the dFSB rescues sleep ontogeny in flies lacking PDM3, demonstrating that greater inhibitory DA signaling to the sleep center prevents young flies from achieving high sleep amounts. Transcriptional profiling of mid-pupal brains and a subsequent genetic modifier screen reveal that *pdm3* regulates expression of the synapse assembly gene, *Msp300*, to control sleep ontogeny. Thus, miswiring during early development leads to a brain structural abnormality that disrupts the normal ontogeny of sleep behavior.

## Results

### *Pdm3* controls sleep ontogeny in *Drosophila*

To identify genes that specifically regulate sleep ontogeny, we performed an RNAi-based screen to search for factors that, when knocked down, abolish normal ontogenetic change in sleep duration. We used Elav-GAL4 to pan-neuronally knockdown >1000 individual genes selected from those with predicted neuronal expression (www.flybase.org), and compared sleep amount in adult flies at day

one post-eclosion (young adults) to flies at day 4–5 post-eclosion (mature adults) (*Figure 1A*). We calculated an 'ontogeny ratio' (OR) as the ratio of sleep amount in young/mature flies of each genotype, and screened for ratios close to 1. After identifying 53 hits from the primary screen (*Figure 1B*; *Figure 1—figure supplement 1A*), we then conducted a secondary screen with independent RNAi lines (*Figure 1C*; *Figure 1—figure supplement 1B*; *Dietzl et al., 2007*).

Knockdown of the gene *pdm3* with either one of two RNAi lines, each targeting a different part of the gene, resulted in a consistent loss of sleep ontogenetic change (*Figure 1B,C*). Following outcrossing to a uniform genetic background, we found that while genetic controls had robust ontogenetic change in total sleep time, pan-neuronal *pdm3* knockdown abolished this transition (*Figure 1D,E*). This phenotype was driven by attenuation of daytime sleep ontogeny (*Figure 1F*) and less night sleep in young compared to mature flies with *pdm3* knockdown (*Figure 1G*). *Pdm3* is part of the POU domain transcription factor family, a gene group with essential roles in nervous system patterning across species (*Badea et al., 2012*; *Dominguez et al., 2013*; *Olsson-Carter and Slack, 2011*). In *Drosophila*, PDM3 is a broadly expressed neuronal protein that coordinates axon targeting in several areas of the brain (*Chen et al., 2012*; *Tichy et al., 2008*). Staining with an anti-PDM3 antibody confirmed a drastic reduction in brain protein levels with knockdown (*Figure 1—figure supplement 2*; *Chen et al., 2012*). Restoring PDM3 in the setting of *pdm3* knockdown rescued sleep ontogeny (*Figure 1H*; *Figure 1—figure supplement 2*), indicating that the phenotype is specific to PDM3 reduction. Thus, *pdm3* regulates sleep ontogenetic change.

Increased daytime sleep in young flies typically stems from prolonged duration of sleep bouts (*Figure 2A*, genetic controls) (*Kayser et al., 2014*); *pdm3* knockdown disrupted this consolidation of daytime sleep in early life (*Figure 2A*). Further, at both ages, flies had more fragmented day and night sleep than controls, with greater numbers of short sleep bouts (*Figure 2A–D*), as well as redistributed sleep across the day and night (*Figure 1D*; *Figure 2—figure supplement 1A*). Mature flies lacking *pdm3* also exhibited reduced locomotor rhythmicity in constant darkness (*Figure 2—figure supplement 1B-D*), though the core molecular clock remained intact (*Figure 2—figure supplement 1E*), suggesting a problem with clock output (*Taghert and Shafer, 2006*). Importantly, these changes in sleep architecture and rhythmicity were fully dissociable from deficits in sleep ontogeny (demonstrated below, *Figure 3E–G*; *Figure 5—figure supplement 1*) indicating a mechanistically distinct role for *pdm3* in juvenile sleep.

The loss of sleep ontogenetic change can arise in two main ways: (1) early life sleep is disrupted, such that young flies do not achieve high sleep amounts or (2) mature flies exhibit a persistent juvenile sleep state. Comparison of sleep duration at each age showed that while mature flies lacking *pdm3* exhibited a mild reduction in sleep duration, young flies had a more substantial loss of sleep (*Figure 2E–H*). Moreover, despite circadian-related redistribution of sleep from night to day, daytime sleep was lower than anticipated in juvenile flies with PDM3 knockdown (*Figure 2—figure supplement 2*). Taken together, our data show that with *pdm3* knockdown, young flies are not able to attain appropriately high sleep amounts or increased sleep consolidation, indicating a particular disruption of early life sleep.

We next wondered whether *pdm3* controls maturation of other behaviors, as opposed to sleep behaviors specifically. Male courtship of females is a robust, innate behavior that also undergoes ontogenetic change during early adulthood (*Eastwood and Burnet, 1977*). In contrast to its effects on sleep ontogeny, *pdm3* knockdown did not affect ontogenetic change in courtship (*Figure 2—figure supplement 3*). Therefore, *pdm3* is unlikely to be broadly required for behavioral maturation, but plays a critical role in regulating sleep ontogeny.

## Pdm3 acts during the mid-pupal stage of development to control sleep ontogeny

*Pdm3* could play an active role in regulating sleep amount in young adult flies, or it could act during earlier development to influence how sleep circuits form. We used an inducible knockdown system (*McGuire et al., 2004*) to examine when during development *pdm3* is required for normal sleep ontogeny. First, we tested whether *pdm3* acts prior to or following eclosion. Beginning knockdown immediately following eclosion had no effect on sleep duration (*Figure 3B*), fragmentation (*Figure 3—figure supplement 1A*), or day-night distribution (*Figure 3—figure supplement 1B*) in mature adults, indicating that *pdm3* does not actively regulate sleep in adulthood. In contrast, knockdown of *pdm3* for the entire pre-eclosion period (starting at the embryonic stage) abolished

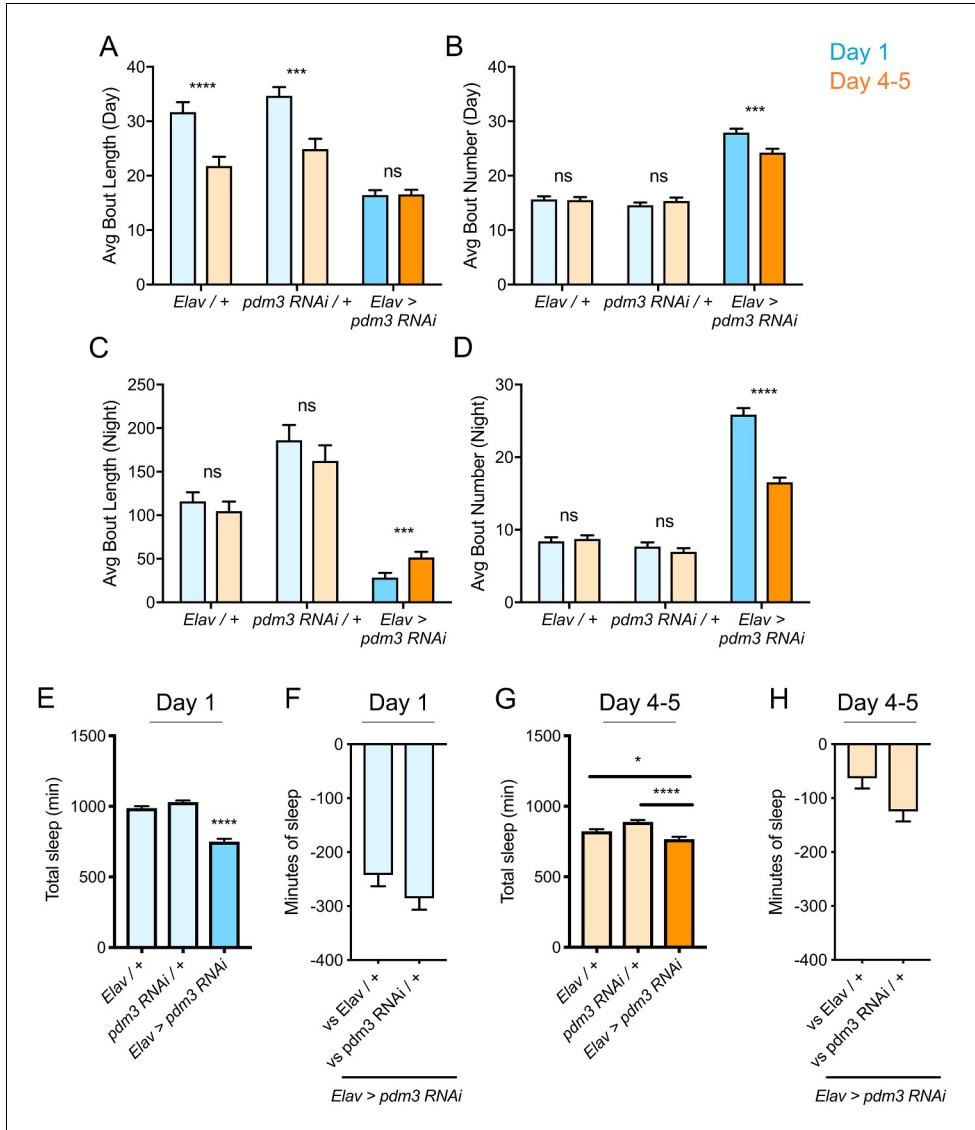

**Figure 2.** Effects of *pdm3* knockdown on sleep architecture. (**A**) Day sleep average bout length and (**B**) day sleep average bout number in *pdm3* RNAi and controls at day 1 (blue) versus day 4–5 (orange). (**C**) Night bout length and (**D**) night average bout number in *pdm3* RNAi and controls (n = 97, 106, 119, 95, 140, 145 left to right in A-D). (**E**) Total sleep time at day 1, *pdm3* RNAi versus controls (n = 97, 119, 140 left to right). (**F**) Minutes of sleep lost in *Elav-GAL4 >UAS pdm3 RNAi* at day one compared to each genetic control. (**G**) Total sleep time at day 4–5, *pdm3* RNAi versus controls (n = 106, 95, 145 left to right). (**H**) Minutes of sleep lost in *Elav-GAL4 >UAS pdm3 RNAi* at day 4–5 compared to each genetic control. ****p<0.0001, ***p<0.001, **p<0.01, *p<0.05; multiple Student's *t* tests with Holm-Sidak correction, alpha = 0.05 (**A–D**); ANOVA with Tukey's test (**E–H**).

The online version of this article includes the following figure supplement(s) for figure 2:

**Figure supplement 1.** *Pdm3* knockdown disrupts behavioral rhythms but leaves molecular clock intact.

**Figure supplement 2.** Comparison of sleep duration within age groups with *pdm3* knockdown.

**Figure supplement 3.** Maturation of courtship behaviors are unaffected by *pdm3* knockdown.

sleep ontogenetic change (*Figure 3C*). We further restricted knockdown to narrower pre-eclosion developmental windows. Loss of *pdm3* from embryonic stages until the beginning of the 3$^{rd}$ instar larval stage had no effect on sleep ontogeny (*Figure 3D*), indicating that *pdm3* acts during later stages of development. Knockdown from the beginning of the 3$^{rd}$ instar larval stage to the middle of pupation likewise did not affect sleep ontogeny (*Figure 3E*, left panel), but did lead to sleep fragmentation (*Figure 3E*, right panel) and day-night redistribution (*Figure 3—figure supplement 1C*).

The disruption of sleep architecture while ontogeny remained intact indicates that these phenotypes are temporally dissociable. Finally, in addition to effects on sleep architecture (*Figure 3F*, right and *Figure 3—figure supplement 1D*), extending knockdown from the beginning of 3rd instar up to the late pupal stage abolished sleep ontogeny (*Figure 3F*, left). Since sleep ontogeny remained intact until knockdown extended through the mid-pupal stage, we concluded that *pdm3* acts during mid-pupal development to control sleep ontogeny (*Figure 3G*).

## *Pdm3* controls dopaminergic synapse formation in the sleep-promoting dorsal fan shaped body

Given that *pdm3* acts during pre-eclosion development, we hypothesized that loss of PDM3 may disrupt patterning of ontogeny-relevant sleep circuits. The dorsal fan shaped body (dFSB), part of the central complex (CCX), is a synaptic neuropil in which projections from wake-promoting DA neurons inhibit sleep-promoting ExFl2 neurons (*Donlea et al., 2011*; *Donlea et al., 2014*; *Liu et al., 2012*; *Pimentel et al., 2016*). These DA inputs are less active in young flies, reducing inhibition of the sleep-promoting neurons and facilitating more sleep (*Kayser et al., 2014*). Previous work showed that *pdm3* null mutants, which do not survive as adults, have grossly disrupted DA innervation of the CCX and intrinsic CCX structural abnormalities (*Chen et al., 2012*). We investigated whether the decrease in juvenile sleep seen with *pdm3* knockdown could be explained by changes in DA innervation of the dFSB, or aberrant development of the CCX itself.

We first examined whether *pdm3* knockdown leads to structural changes in the sleep-promoting ExFl2 projections to the dFSB, labeled by the 23E10 driver (*Donlea et al., 2014*). We did not see gross morphological changes in the neurites (*Figure 4A*), but quantification revealed an increase in innervation with *pdm3* knockdown (*Figure 4B*). An increase in ExFl2 projections to the dFSB could reflect the sleep-promoting neurons receiving more inhibitory DA input, thereby preventing young flies from achieving high sleep amounts. Indeed, labeling of DA neurites using a tyrosine hydroxylase promoter (TH-LexA) showed a two-fold increase in the density of DA innervation to the dFSB (*Figure 4C,D*). We also noticed that DA inputs to the ventral FSB (vFSB) were grossly disorganized (*Figure 4C*). However, loss of PDM3 did not diffusely disrupt DA innervation, as TH+ innervation of the mushroom body was unchanged (*Figure 4—figure supplement 1A*). Thus, *pdm3* knockdown alters DA inputs to the FSB, leading to greater innervation density in the dFSB and disorganization of inputs to the vFSB.

We next wanted to determine whether the increased density of DA-dFSB neurites reflects more pre-synaptic sites, as these projections may not actually form synapses. We expressed a fluorescently tagged form of the pre-synaptic active zone protein, Bruchpilot (Brp-short^mCherry) (*Fouquet et al., 2009*; *Wagh et al., 2006*) specifically in TH+ neurons (*Figure 4E*; *Figure 4—figure supplement 1B*). In control flies, the density of DA synapses to the dFSB increased as flies matured (*Figure 4F*). Thus, DA inputs to the dFSB are both more active (*Kayser et al., 2014*) and more numerous in mature flies. By contrast, in a young fly lacking *pdm3*, the number of DA synapses to the dFSB was elevated (*Figure 4F*), and ontogenetic changes were abolished (*Figure 4F*). In the vFSB, *pdm3* knockdown led to a reduction in TH+ synapses at both ages, and ontogenetic change in synapse number was still preserved (*Figure 4—figure supplement 1C*). Thus, with *pdm3* knockdown, we observed a particular disruption to developmentally regulated synapse addition in the dFSB but not vFSB. The number of 23E10+ and TH+ neurons projecting to the FSB was unchanged with *pdm3* knockdown when compared to controls, indicating that *pdm3* does not affect early cell fate decisions in this circuit (*Figure 4—figure supplement 1D-F*). Together, these data suggest that a premature increase in DA synapses to the dFSB underlies loss of behavioral sleep ontogeny.

## Reducing dopaminergic signaling in the dFSB rescues sleep ontogenetic change

We hypothesized that with *pdm3* knockdown, greater DA synaptic input increases inhibition of 23E10+ dFSB neurons, preventing high dFSB activity normally seen in young flies. To measure 23E10+ neuron activity, we used the CaLexA (Calcium-dependent nuclear import of LexA) system (*Masuyama et al., 2012*), which relies on activity-dependent nuclear import of a chimeric transcription factor to drive GFP expression. We generated *pdm3* RNAi in the QUAS system for compatibility with CaLexA reagents (*Figure 4—figure supplement 2*), and found *pdm3* knockdown led to a

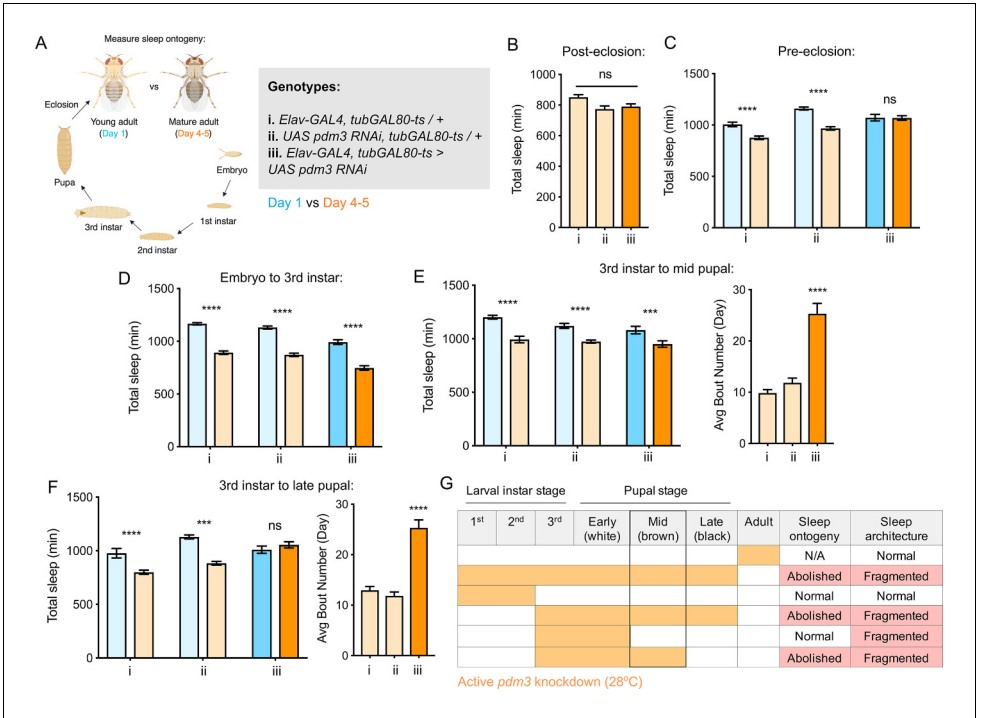

**Figure 3.** *Pdm3* acts during mid-pupal development to control sleep ontogeny.  (**A**) *Drosophila* life cycle. (**B**) Total sleep in mature adults with *pdm3* knockdown post-eclosion and genetic controls (n = 50, 60, 70 left to right). (**C**) Total sleep with pre-eclosion *pdm3* knockdown in day 1 (blue) versus day 4–5 (orange) (n = 44, 42, 38, 42, 35, 50 left to right) (**D**) Total sleep with *pdm3* knockdown from embryo to the 3$^{rd}$ instar larval stage (n = 76, 79, 82, 64, 73, 99 left to right). (**E**) Total sleep (left) and day sleep bout number (right) with *pdm3* knockdown from the 3$^{rd}$ instar larval stage up to the mid pupal stage (n = 33, 32, 32, 31, 32, 32 left to right). (**F**) Total sleep (left) and day sleep bout number (right) with *pdm3* knockdown from the 3$^{rd}$ instar larval stage to the late pupal stage (n = 30, 32, 32, 32, 32, 31 left to right). (**G**) Summary of temporal mapping and dissociation of sleep ontogeny from sleep architecture ****p<0.0001, ***p<0.001, **p<0.01, *p<0.05; multiple Student's *t* tests with Holm-Sidak correction, alpha = 0.05 (C, D, E/F left); ANOVA with Tukey's test (B, E/F right).

The online version of this article includes the following figure supplement(s) for figure 3:

**Figure supplement 1.** Sleep architecture with temporally-restricted *pdm3* knockdown.

---

decrease in the 23E10+ dFSB CaLexA signal in day one flies (***Figure 4G,H***). Thus, increased TH+ innervation of the dFSB reduces activity of 23E10+ sleep-promoting neurons in young flies.

If DA-driven inhibition of 23E10+ neurons underlies the lack of sleep ontogenetic change with *pdm3* knockdown, blocking DA signal transmission should rescue sleep ontogeny. DA signaling promotes wake by inhibiting sleep-promoting 23E10+ neurons via two D1-like receptors, Dop1R1 and Dop1R2 (***Liu et al., 2012***; ***Pimentel et al., 2016***; ***Ueno et al., 2012***). We knocked down *pdm3* pan-neuronally in a *Dop1R1* and *Dop1R2* double null mutant background (hereafter referred to as *Dop1R1/R2 -/-*)(***Keleman et al., 2012***). *Dop1R1/R2 -/-* mutants exhibited increased sleep amounts overall and retained normal sleep ontogenetic change (***Figure 5A***, genotypes i and ii). In keeping with the reported haplosufficiency of this mutant, knockdown of *pdm3* in a *Dop1R1/R2 +/-* background mimicked knockdown in an otherwise wild-type background, showing loss of sleep ontogenetic change (***Figure 5A,iii***). However, complete loss of these receptors (*Dop1R1/R2 -/-*) rescued sleep ontogeny in the setting of *pdm3* knockdown (***Figure 5A***, iv).

The *Dop1R1/R2 -/-* mutant did not rescue sleep fragmentation, day-night sleep redistribution, or circadian arrhythmicity, demonstrating again that these phenotypes are mechanistically independent from loss of sleep ontogenetic change with *pdm3* knockdown (***Figure 5—figure supplement 1A-C***). Further, comparison of sleep amounts at each age in the setting of *pdm3* knockdown showed that *Dop1R1/2 -/-* specifically increases sleep in young flies (***Figure 5B***), but does not alter mature sleep amounts (***Figure 5C***). Thus, loss of the two dFSB DA receptors, Dop1R1 and Dop1R2, specifically

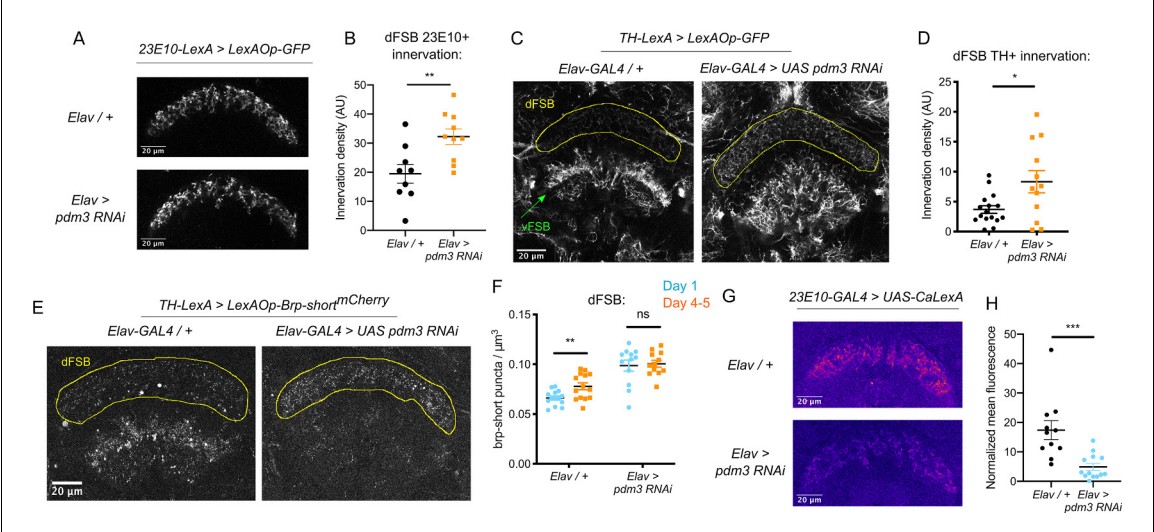

**Figure 4.** Loss of PDM3 increases inhibitory DA input to the sleep-promoting dFSB. (**A**) Projections of 23E10+ dFSB neurons in controls (top) and *pdm3* RNAi (bottom). (**B**) Innervation density of 23E10+ neurites in the adult dFSB (n = 9 controls, 10 *pdm3* RNAi). (**C**) TH+ projections to the FSB in controls (left) and *pdm3* RNAi (right). (**D**) Innervation density of TH+ neurites in the adult dFSB (n = 16 controls, 12 *pdm3* RNAi). (**E**) Labeling of TH+ pre-synaptic sites in the FSB with Brp-short^mCherry. (**F**) TH+ synapse density in the dFSB in controls (left) and *pdm3* RNAi (right) at day 1 and day 4–5 (n = 14, 14, 12, 12 left to right). (**G**) Pseudocolored CaLexA signal in 23E10+ neurons in day one controls (top) versus pdm3 RNAi (bottom), quantified in (**H**) (n = 11 controls, 13 *pdm3 RNAi*). ****p<0.0001, ***p<0.001, **p<0.01, *p<0.05; unpaired two-tailed Student's *t* test plus Welch's correction (**B,D,H**), multiple Student's *t* tests with Holm-Sidak correction, alpha = 0.05 (**F**).

The online version of this article includes the following figure supplement(s) for figure 4:

**Figure supplement 1.** Additional characterization of dopaminergic innervation and cell counts in the setting of *pdm3* knockdown.

**Figure supplement 2.** Confirmation of QF2-QUAS system for panneuronal *pdm3* knockdown.

restores sleep ontogenetic change in a manner dissociable from other *pdm3*-related sleep behaviors.

Lastly, to test the prediction that the loss of sleep ontogeny stems from heightened DA signaling specifically through the dFSB, we knocked down *Dop1R1* using *23E10-GAL4* in the setting of *pdm3* RNAi. Indeed, reduction of dopaminergic influence in the dFSB rescued sleep ontogenetic change (*Figure 5D*) but, like the null mutant, did not affect sleep architecture (*Figure 5—figure supplement 1D*). Together, these results indicate that a developmental wiring error in flies lacking PDM3 prematurely increases wake-promoting DA-dFSB signaling in young adult flies, abolishing juvenile sleep.

## *Pdm3* acts in primordial central complex neurons to influence TH+ dFSB innervation

We next sought to determine which cell population *pdm3* acts in to control sleep ontogeny, and how this wiring error occurs. *Pdm3* could either act in pre-synaptic TH+ cells or in CCX neurons that are targeted by TH+ inputs. Knockdown of *pdm3* using TH-GAL4, which targets the majority of DA neurons and is expressed in the CCX-projecting DA neurons during pupal development (*Hartenstein et al., 2017*), did not affect sleep ontogeny (*Figure 6A*). *Pdm3* knockdown using 23E10-GAL4 also did not affect sleep ontogeny (*Figure 6A*); however, during the mid-pupal stage when *pdm3* is acting, 23E10-GAL4 is not expressed in neurons that project to the FSB (*Figure 6—figure supplement 1*). To address whether *pdm3* acts in neurons that are targets of TH+ inputs, we needed to test GAL4 drivers that express in CCX neurons at the mid-pupal developmental stage. We reasoned that at least some adult-defined drivers might have similar expression patterns in the pupal brain, and screened a subset of GAL4 lines with adult CCX expression from the FlyLight collection (*Figure 6B*; *Figure 6—figure supplement 2A*; *Jenett et al., 2012*). We identified one driver, R93F07-GAL4, which produced a loss of sleep ontogenetic change (*Figure 6B,C*) and was expressed in the CCX during pupal development. During the mid-pupal stage, R93F07-GAL4 targets both the primordial FSB and ellipsoid body (*Figure 6D*). These R93F07+ neurons are not TH+ (*Figure 6—*

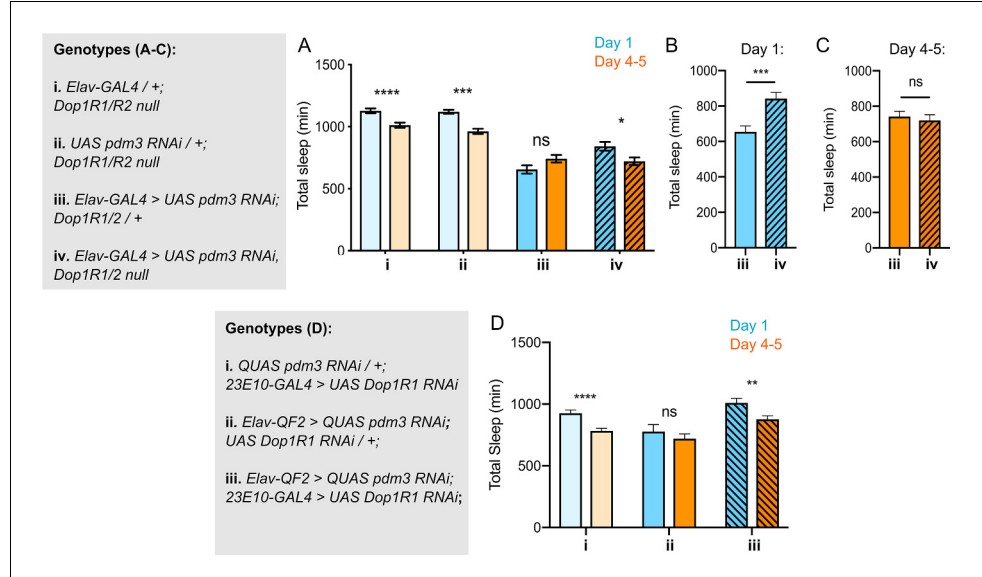

**Figure 5.** Reducing Dop1R1 signaling restores normal sleep ontogenetic change. (**A**) *Dop1R1/R2-/-* with *pdm3* RNAi and controls (n = 55, 92, 72, 96, 58, 97, 45, 75 left to right). Comparison of total sleep amount in Elav-pdm3 RNAi with and without Dop1R1/R2 null mutation in day 1 (**B**) and day 4–5 (**C**). (**D**) *Dop1R1* knockdown in 23E10+ neurons in the setting of pan-neuronal *pdm3* RNAi and controls (n = 39, 46, 17, 25, 33, 50 left to right). ****$p<0.0001$, ***$p<0.001$, **$p<0.01$, *$p<0.05$; multiple Student's *t* tests with Holm-Sidak correction, alpha = 0.05 (**A, D**), unpaired two-tailed Student's *t* test plus Welch's correction (**B,C**).

The online version of this article includes the following figure supplement(s) for figure 5:

**Figure supplement 1.** Reducing dopaminergic signaling does not rescue aberrant sleep architecture.

*figure supplement 2B*), further supporting the idea that *pdm3* is not acting in TH+ neurons themselves. Importantly, knockdown of *pdm3* using other developmentally expressed sleep/circadian drivers did not affect sleep ontogeny (*Figure 6A*). Additionally, glial-specific knockdown did not affect sleep ontogeny, confirming that our phenotype stems from loss of PDM3 in neurons (*Figure 6A*). Thus, *pdm3* appears to act in primordial CCX target neurons to coordinate DA innervation of the dFSB. To test this directly, we knocked down *pdm3* using R93F07-GAL4 and measured TH+ innervation density in the dFSB. Indeed, restricting *pdm3* knockdown to TH-, R93F07+ CCX neurons increased the density of DA innervation in the dFSB (*Figure 6E*), confirming that *pdm3* acts in CCX target neurons to control DA innervation of this region.

How does DA innervation of the dFSB go awry during development? Little is known about when and how DA inputs normally integrate into CCX circuitry. To characterize this process, we tracked the ingrowth of TH+ neurites to the FSB during pupation. In controls at 24 hr after puparium formation (APF), TH+ neurites were rarely observed in the dFSB, but we observed an early phase of TH+ innervation in the vFSB (*Figure 7A*, upper left and *Figure 7B/C*, left). By 48 hr APF, DA innervation in both the dFSB and vFSB closely resembled the adult brain (*Figure 7A*, lower left). By contrast, *pdm3* knockdown led to a striking increase in TH+ innervation of the dFSB at 24 hr APF, while vFSB innervation proceeded unchanged (*Figure 7A*, upper right and *Figure 7B/C*, right). As with controls, TH+ innervation in the dFSB and vFSB with *pdm3* knockdown resembled adult brains at 48 hr APF (*Figure 7A*, lower right). Thus, TH+ neurites normally innervate the vFSB first, and by 48 hr APF, layered TH+ innervation of the entire FSB appears largely complete. Loss of PDM3 leads to early ingrowth of TH+ neurites to the dFSB, likely increasing the amount of dFSB TH+ innervation in the adult brain.

### *Pdm3* regulates the synaptic gene, *Msp300*, to control sleep ontogeny

How does *pdm3* coordinate appropriate DA innervation of the dFSB at the molecular level? *Pdm3* binds to a TAAT Hox motif 2–3 base pairs upstream of a POU motif (TGCAA/T) (*Andersen and Rosenfeld, 2001*; *Jafari and Alenius, 2015*). Transcriptional targets of *pdm3* remain largely

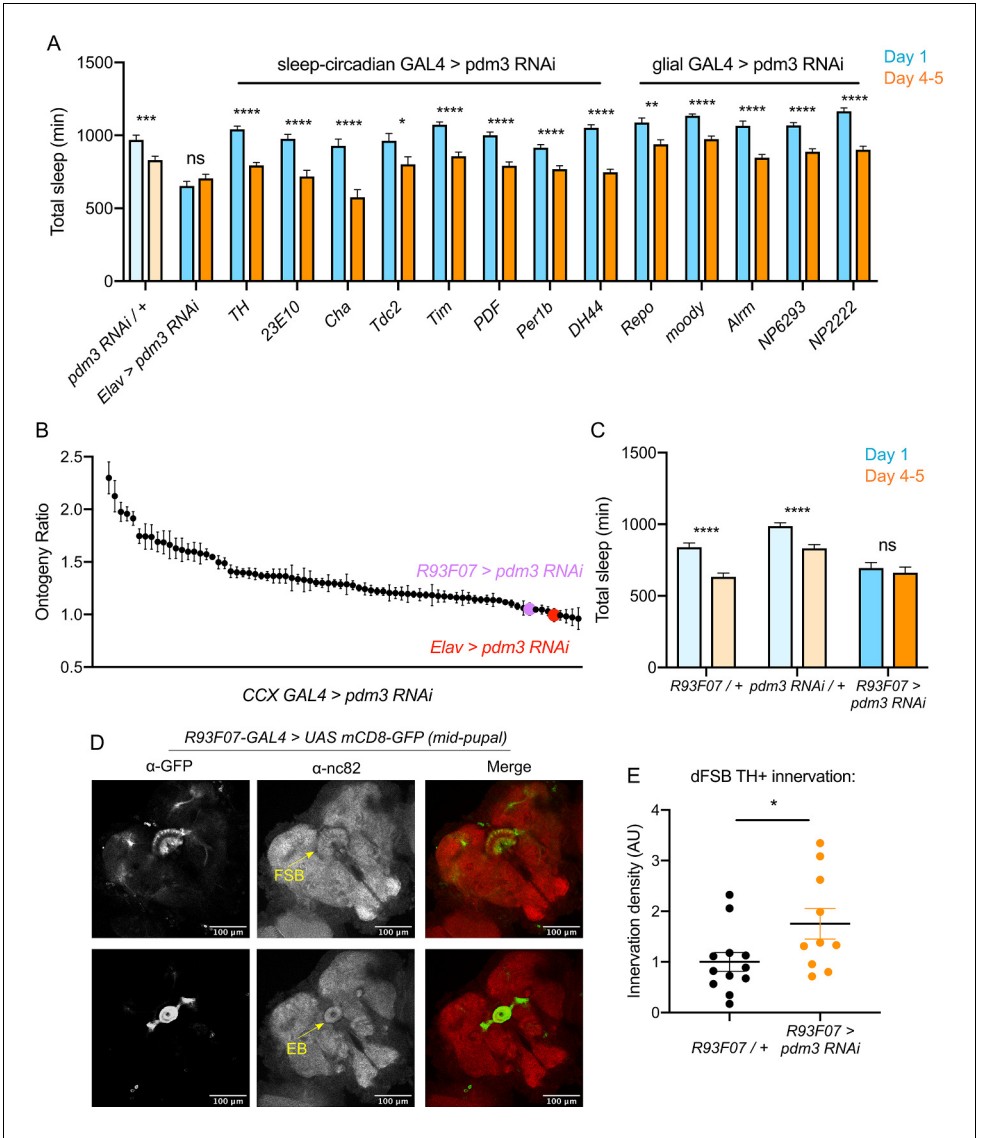

**Figure 6.** Pdm3 acts in R93F07+ CCX target cells to control sleep ontogeny. (**A**) Total sleep time: *pdm3* knockdown with spatially restricted GAL4 drivers with expression in sleep/circadian circuitry or glial expression (n ≥ 8 per genotype/age). (**B**) Spatial mapping screen of FlyLight GAL4 lines with adult CCX expression patterns (n ≥ 8 per genotype/age). (**C**) Total sleep time in *R93F07-GAL4 > UAS pdm3 RNAi* versus controls (n = 41, 32, 40, 40, 40, 40 left to right). (**D**) At the mid-pupal stage, R93F07-GAL4 is expressed in the FSB (top) as well as the ellipsoid body (EB, bottom). (**E**) Innervation density of TH+ neurites in the adult dFSB (labeled by TH-LexA >LexAOp GFP) with R93F07-GAL4 driving *pdm3* RNAi (n = 12 controls, 10 *pdm3 RNAi*). ****p<0.0001, ***p<0.001, **p<0.01, *p<0.05; multiple Student's *t* tests with Holm-Sidak correction, alpha = 0.05 (**A, C**), unpaired two-tailed Student's *t* test plus Welch's correction (**E**).

The online version of this article includes the following figure supplement(s) for figure 6:

**Figure supplement 1.** Expression pattern of 23E10-GAL4 in the mid-pupal brain.

**Figure supplement 2.** Sleep amounts from spatial mapping screen identifying R93F07+ cells.

uncharacterized (*Jafari and Alenius, 2015*; *Tichy et al., 2008*). To determine which genes affect sleep ontogeny downstream of *pdm3*, we performed RNA-Seq analysis of central brains in *pdm3* knockdown and controls at the mid-pupal stage (*Figure 8A*, n = 4 biological replicates). Staining of pupal brains confirmed *pdm3* knockdown (*Figure 8—figure supplement 1*). Differential gene expression analysis revealed 35 genes that were downregulated with *pdm3* knockdown and 54 genes that were upregulated (p-adj <0.05, Fold Change > 1.2; *Supplementary file 1*; *Figure 8B*).

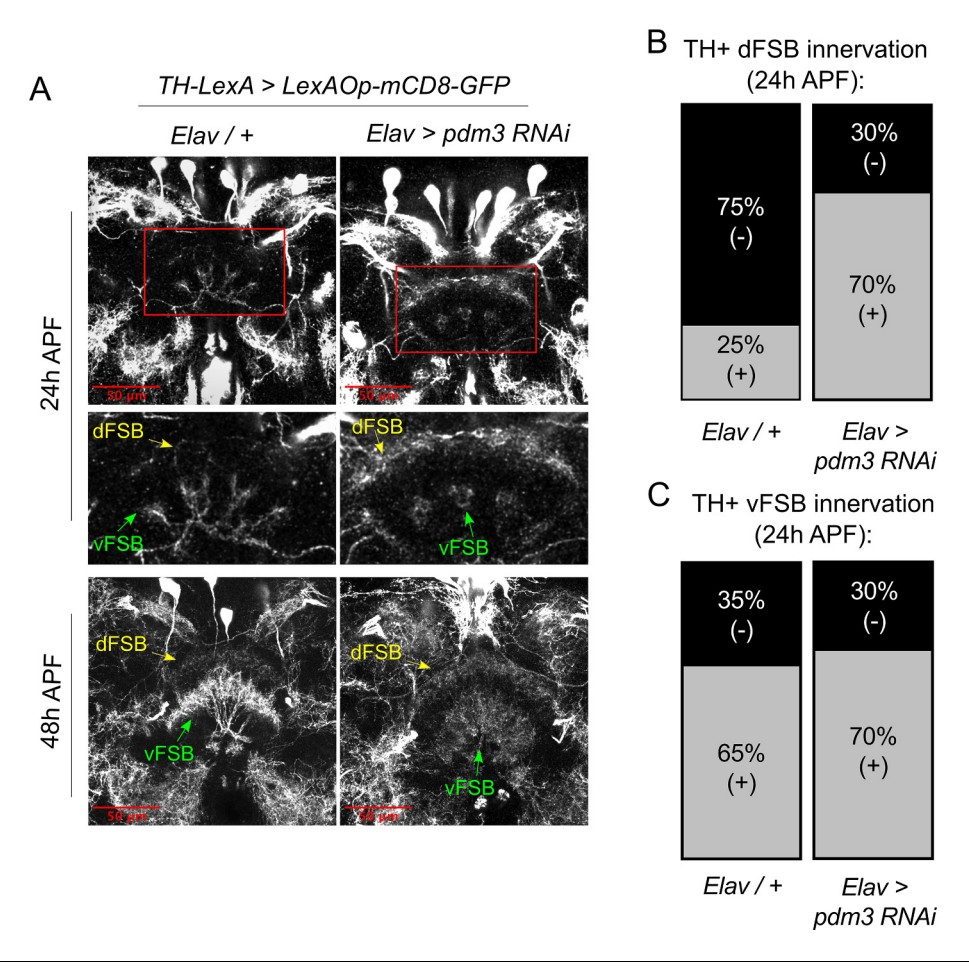

**Figure 7.** *Pdm3* controls TH+ dFSB innervation during pupation. (**A**) TH+ innervation in the FSB at 24 hr after puparium formation (APF) (top) and 48 hr APF (bottom). (**B,C**) Percentage of brains with TH+ innervation in the dFSB (**B**) or vFSB (**C**) at 24 hr APF (gray = positive for innervation, black = negative; n = 10 controls, 10 *pdm3* RNAi).

Interestingly, some of the most significant changes in expression occurred in genes known to facilitate synapse formation and plasticity, such as *Arc1* and *Msp300* (*Figure 8B*; *Ashley et al., 2018*; *Morel et al., 2014*; *Packard et al., 2015*). Since loss of *pdm3* results in excessive DA synaptic ingrowth in the dFSB, we hypothesized that *pdm3* may regulate synapse dynamics by directing expression of some of these genes in particular.

To identify transcriptional changes relevant to the loss of sleep ontogeny, we performed a modifier RNAi screen (*Figure 8A*). We focused specifically on the top genes that were upregulated, and combined RNAi against these genes with *pdm3* RNAi to test if reducing their levels rescued the sleep ontogeny phenotype. Expression of *pdm3* RNAi along with *UAS-Dcr2* (to facilitate knockdown of other genes) resulted in a more drastic sleep ontogeny phenotype, with young flies sleeping less than mature flies and an ontogeny ratio below 1.0 (*Figure 8C*; *Figure 8—figure supplement 2A*). Co-expression of most RNAi lines did not result in a significant rescue of sleep ontogeny (*Figure 8C*). However, in the setting of *pdm3* RNAi, knockdown of *Msp300* using three independent RNAi lines suppressed the *pdm3* sleep ontogeny phenotype (*Figure 8C*; *Figure 8—figure supplement 2A,B*) without affecting PDM3 knockdown level (*Figure 8—figure supplement 2C*). Expression of *Msp300* RNAi alone had no effect on baseline sleep or sleep ontogeny, nor did *UAS-Dcr2* itself (*Figure 8—figure supplement 2D*).

To determine whether PDM3 directly binds and regulates *Msp300*, we took advantage of a preexisting ChIP-Seq dataset available at the ENCODE database (www.encodeproject.org)

(*Davis et al., 2018*). ChIP-Seq for *pdm3* was performed at the pre-pupal stage, close the critical time window for *pdm3* regulation of sleep ontogeny. Comparison of the top 200 genes by p-value and signal strength in this dataset to differentially expressed genes from our RNA-Seq data showed *Msp300* is one of five genes present in both datasets (*Figure 8D*). Visualization of *Msp300*-associated peaks revealed a strong binding peak across all replicates located approximately 400 bp upstream from the transcription start site in the putative *Msp300* promoter region of three *Msp300* isoforms, as well as an intronic peak (*Figure 8E*). Our RNA-Seq analysis with *pdm3* knockdown demonstrated increased reads across exons specific to the *Msp300-RD, RL* and *RB* isoforms compared to control, immediately downstream of the strongest binding peak (*Figure 8E*; *Figure 8—figure supplement 3A*, shaded boxes), suggesting that *pdm3* may regulate these *Msp300* isoforms. Thus, the observed changes in *Msp300* mRNA level with *pdm3* knockdown are likely due to direct transcriptional regulation.

Finally, in the setting of *pdm3* RNAi, we tested whether *Msp300* knockdown also rescued excessive TH+ dFSB innervation. Indeed, TH+ dFSB innervation with *pdm3* knockdown was significantly reduced upon co-expression of *Msp300* RNAi (*Figure 8F*), further supporting the idea that exaggerated TH+ input to the dFSB drives the loss of sleep ontogeny. Our model predicts that *pdm3* and *Msp300* act within the same population of CCX target neurons defined by *R93F07*-GAL4. To test this idea directly, we knocked down PDM3 expression pan-neuronally while expressing *Msp300* RNAi with R93F07-GAL4. Surprisingly, this manipulation did not suppress the *pdm3*-related sleep ontogeny phenotype (*Figure 8—figure supplement 3B*), suggesting a more complex interaction between *pdm3* and its downstream transcriptional targets. Regardless, these results indicate *pdm3* acts through suppression of *Msp300* levels during pupal development to control wiring of sleep ontogeny circuitry.

## Discussion

Despite a long-standing appreciation for the importance of early life sleep in brain development, its regulation at the molecular level has thus far been unknown. We have identified the transcription factor, *pdm3*, as a genetic regulator of sleep ontogeny. *Pdm3* acts during early development to coordinate synapse formation in a sleep ontogeny circuit, enabling the brain to later execute high levels of sleep. These findings provide novel insight into how a central component of *Drosophila* sleep circuitry is wired, and how disruption of this process impinges on sleep behavior.

At the circuit level, *pdm3* knockdown leads to premature and exaggerated DA input to the dFSB, inhibiting sleep output neurons. Rather than acting in TH+ neurons themselves, *pdm3* functions within CCX neurons (labeled by *R93F07-GAL4*) to pattern DA inputs in this region. CCX intrinsic neurons form rudimentary projections to the FSB neuropil during larval development (*Riebli et al., 2013*), prior to DA innervation, supporting the idea that DA inputs are directed by pre-existing CCX targets. How does early entrance of TH+ neurites into dFSB territory lead to the eventual increase in innervation in the adult brain? We propose that *pdm3* restricts the timing of a developmental window during which target-derived cues facilitate the guidance and stabilization of TH+ processes in the dFSB. Upon PDM3 reduction, this window opens early, allowing more time for innervation and an increase in TH+ synapse number. Our data suggest *pdm3* controls development of this circuit and behavioral sleep ontogeny via repression of *Msp300*. *Msp300*, also known as dNesprin-1, is a cytoskeleton-associated protein that facilitates delivery of mRNAs to promote synapse maturation (*Morel et al., 2014*; *Packard et al., 2015*). As such, an increase in Msp300 may result in exaggerated or inappropriately-timed delivery of synaptic mRNAs and exuberant synaptogenesis. Restriction of Msp300 knockdown to R93F07+ cells failed to rescue the loss of sleep ontogeny observed with pan-neuronal PDM3 depletion. This result raises the possibility of mixed cell-autonomous and non-cell-autonomous interactions between these molecules in coordinating DA innervation of the CCX, and/or a role for other PDM3 transcriptional targets. Future studies will further delineate how *pdm3* coordinates expression of *Msp300* and other synaptic genes to orchestrate development of this sleep circuit.

POU family proteins have highly conserved roles in nervous system patterning across species. In mammals, POU proteins have been linked to neurodevelopmental diseases that often have comorbid sleep abnormalities (*Kotagal, 2015*; *Robinson-Shelton and Malow, 2016*) hinting at potential early life sleep-regulatory roles. In particular, the human homolog of *pdm3*, *POU6F2,* has been

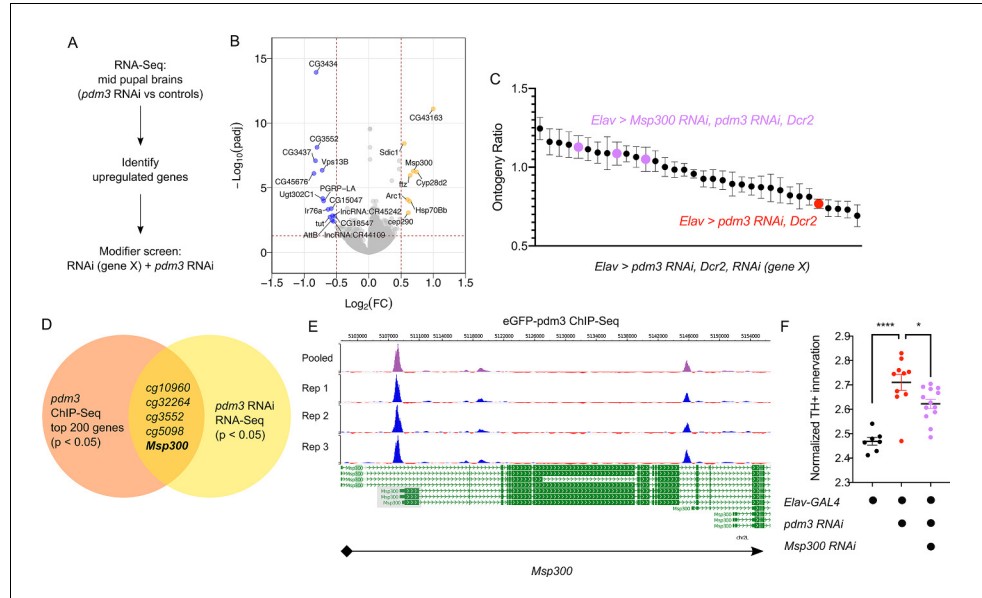

**Figure 8.** *Pdm3* controls expression of the synaptic gene *Msp300* to regulate sleep ontogeny. (**A**) Scheme of RNA-Seq and subsequent modifier screen. (**B**) Volcano plot of RNA-Seq data highlighting significant changes in gene expression with *pdm3* knockdown compared to controls in mid-pupal brains. Labeled genes have $-\log_{10}(padj) > 1.3$ and absolute value of $\log_2$(Fold Change)>0.5. Yellow = increased expression, blue = decreased expression upon *pdm3* knockdown (n = 4 biological replicates per genotype, 40 brains per replicate for RNA-Seq). (**C**) Modifier screen with co-expression of RNAi targeting upregulated genes from RNA-Seq alongside *pdm3* RNAi (n ≥ 16 per genotype/age). (**D**) Overlap of hits from RNA-Seq and *pdm3* ENCODE ChIP-Seq experiments. (**E**) Control-normalized peaks of PDM3 binding within the *Msp300* gene. The strongest binding peak occurs upstream of the first exon in the RD, RL and RB transcript isoforms (shaded gray box). (**F**) Quantification of TH+ staining in the dFSB with *Msp300* RNAi and *pdm3* RNAi (n = 7, 10, 13 left to right). RNA-Seq statistical analysis is detailed in Materials and methods. ****p<0.0001, *p<0.05; ANOVA with Tukey's test (**F**).

The online version of this article includes the following figure supplement(s) for figure 8:

**Figure supplement 1.** Confirmation of PDM3 protein reduction at the mid-pupal developmental timepoint used for RNA-Seq experiments.

**Figure supplement 2.** Additional modifier screen data and further characterization of *Msp300* phenotype.

**Figure supplement 3.** Molecular and cellular interactions between Msp300 and PDM3.

associated with subtypes of autism spectrum disorder (ASD) (*Anney et al., 2010*). Additionally, disrupted function of the *Brn-2* gene (also known as *POU3F2*) is associated with ASD in humans and ASD-like social deficits in mice (*Belinson et al., 2016*; *Marchetto et al., 2017*; *Salyakina et al., 2011*). Developmental sleep abnormalities could thus contribute to behavioral pathology seen with these lesions.

Given the cross-species conservation and importance of sleep ontogeny for brain development, there are likely to be additional genes that control this behavior. In particular, there may be molecules that actively regulate sleep in young flies or genetic lesions that result in a persistent juvenile sleep state. Notably, the developmental role of *pdm3* stands in contrast to most other known *Drosophila* sleep genes (*Chakravarti et al., 2017*; *Dubowy and Sehgal, 2017*) and suggests the existence of an entirely separate class of 'sleep genes' that orchestrate establishment of sleep circuits. This idea raises the intriguing possibility that primary sleep disorders such as insomnia or hypersomnia may have neurodevelopmental origins. The identification of genes and circuits regulating sleep ontogeny thus deepens our understanding of how sleep matures and its age-specific functions for the nervous system.

# Materials and methods

## Fly stocks

Key resources table

| Reagent type (species) or resource | Designation | Source or reference | Identifiers | Additional information |
|---|---|---|---|---|
| Gene *Drosophila melanogaster* | pdm3 | | FBgn0261588 | |
| Gene *Drosophila melanogaster* | Msp300 | | FBgn0261836 | |
| Genetic reagent (*D. melanogaster*) | hs-hid; Elav-GAL4; UAS Dcr2 | Dragana Rogulja | | |
| Genetic reagent (*D. melanogaster*) | UAS-pdm3 RNAi | Bloomington *Drosophila* Stock Center | BSC #53887, TRiP HMJ21205 | |
| Genetic reagent (*D. melanogaster*) | UAS-pdm3 RNAi | Vienna *Drosophila* Resource Center | VDRC #30538, Construct ID 4312 | |
| Genetic reagent (*D. melanogaster*) | ElavC155-GAL4 | Amita Sehgal | | |
| Genetic reagent (*D. melanogaster*) | UAS-mCD8-GFP | Bloomington *Drosophila* Stock Center | | |
| Genetic reagent (*D. melanogaster*) | UAS-pdm3-short | John Carlson | | |
| Genetic reagent (*D. melanogaster*) | tubGAL80-ts | Bloomington *Drosophila* Stock Center | BSC #7017 | |
| Genetic reagent (*D. melanogaster*) | R23E10-LexA | Bloomington *Drosophila* Stock Center | BSC #52693 | |
| Genetic reagent (*D. melanogaster*) | TH-LexA | Ronald Davis | | |
| Genetic reagent (*D. melanogaster*) | LexAOp-GFP | Bloomington *Drosophila* Stock Center | BSC #32203 | |
| Genetic reagent (*D. melanogaster*) | LexAOp-Brp-short-mCherry | Takashi Suzuki | | |
| Genetic reagent (*D. melanogaster*) | 23E10-GAL4 | Bloomington *Drosophila* Stock Center | BSC #49032 | |
| Genetic reagent (*D. melanogaster*) | UAS-CaLexA | J. Wang | | |
| Genetic reagent (*D. melanogaster*) | ElavC155-QF2 | Bloomington *Drosophila* Stock Center | BSC #66466 | |
| Genetic reagent (*D. melanogaster*) | QUAS-pdm3 RNAi | This study | | *Figure 4—figure supplement 2*; available upon request |
| Genetic reagent (*D. melanogaster*) | Dop1R1[attp] | Krystyna Keleman | | |
| Genetic reagent (*D. melanogaster*) | *UAS-Dcr2* | Bloomington *Drosophila* Stock Center | BSC #24651 | |
| Genetic reagent (*D. melanogaster*) | *UAS-Msp300 RNAi* | Vienna *Drosophila* Resource Center | VDRC #105694 | |
| Genetic reagent (*D. melanogaster*) | *UAS-Msp300 RNAi* | Vienna *Drosophila* Resource Center | VDRC #109023 | |
| Genetic reagent (*D. melanogaster*) | *UAS-Msp300 RNAi* | Vienna *Drosophila* Resource Center | VDRC #107183 | |
| Antibody | anti-PDM3 (Guinea pig polyclonal) | Cheng-Ting Chien | | 1:500 |
| Antibody | anti-GFP (Mouse monoclonal) | Sigma | Cat# G6539-200UL | 1:500 |

*Continued on next page*

*Continued*

| Reagent type (species) or resource | Designation | Source or reference | Identifiers | Additional information |
|---|---|---|---|---|
| Antibody | anti-GFP (Rabbit polyclonal) | Fisher | Cat# A11122 | 1:500 |
| Antibody | anti-PER (Guinea pig polyclonal) | Amita Sehgal | | 1:1000 |
| Antibody | anti-PDF C7 (Mouse monoclonal) | Developmental Studies Hybridoma Bank | | 1:500 |
| Antibody | anti-nc82 (Brp) (Mouse monoclonal) | Developmental Studies Hybridoma Bank | | 1:100 |
| Antibody | Alexa Fluor 488 Donkey anti-Rabbit | Thermo Fisher | | 1:1000 |
| Antibody | Alexa Fluor 555 Donkey anti-GP | Thermo Fisher | | 1:1000 |
| Antibody | Alexa Fluor 488 Donkey anti-Mouse | Thermo Fisher | | 1:1000 |
| Antibody | Alexa Fluor 647 Donkey anti-Mouse | Thermo Fisher | | 1:1000 |
| Recombinant DNA reagent | QUAS-WALIUM20 (vector) | Jonathan Zirin, Fly Transgenic RNAi Project | | |
| Sequence-based reagent | HMJ21205 pdm3 RNAi hairpin | Synthesized in this study, sequence from TRiP database | | |
| Restriction enzyme | NheI | New England Biolabs | Cat. #: R3131S | |
| Restriction enzyme | EcoRI | New England Biolabs | Cat # R3101L | |
| Commercial assay or kit | Plasmid Midi Kit | Qiagen | Cat # 12143 | |
| Software, algorithm | Prism 8 | Prism | | |

## Sleep assays

Flies were raised and maintained in bottles on cornmeal molasses food obtained from Lab Express (Fly Food R, recipe available at http://lab-express.com/DIS58.pdf) at 25°C on a 12 hr:12 hr LD cycle. For all ontogeny experiments unless otherwise noted, day one females (young adults) were compared to day 4–5 females (mature adults) of the same genotype. For day 4–5 flies, newly eclosed females were collected and aged in group housing on standard food at 25 degrees on a 12 hr:12 hr LD cycle, and flipped onto new food every 3–4 days. For experiments with young flies, newly eclosed females were group housed until loading into the sleep experiment. Flies were anesthetized on $CO_2$ pads (Genesee Scientific Cat #59–114) and loaded into individual glass tubes (containing 5% sucrose and 2% agar) for monitoring locomotor activity in the *Drosophila* Activity Monitoring (DAM) system (Trikinetics, Waltham MA). All sleep experiments were loaded between ZT5-ZT7, and data collection began at ZT0 the following day (at least 16 hr after $CO_2$ exposure). Activity was measured in 1 min bins and sleep was identified as 5 min of consolidated inactivity (*Hendricks et al., 2000*; *Shaw et al., 2000*). Data was processed using PySolo (*Gilestro and Cirelli, 2009*).

## Ontogeny screen

For the sleep ontogeny primary and secondary screens, the hs-hid; Elav-GAL4, UAS Dcr2 (hEGD) fly stock was crossed to each RNAi line from the TRiP, GD and KK collections. We screened genes with neuronal expression (www.flybase.org, 1067 genes). For each gene, we screened one RNAi line from the Transgenic RNAi Project (TRiP) collection (*Ni et al., 2009*). Control lines for the primary screen were hEGD x TRiP library landing site host strains: P{y[+t7.7]=CaryP}attP2 (Chr3, BSC #36303) or P{y[+t7.7]=CaryP}attP40 (Chr2, BSC #36304). Controls for the secondary screen were hEGD x Landing site VIE-260B KK library host strain (VDRC #60100) and GD library host strain (VDRC #60000). Day one females were loaded alongside day 4–5 females of the same genotype. Since young flies

typically sleep about twice as much during the day compared to mature flies (*Kayser et al., 2014*), we focused on daytime sleep for our screen. An 'ontogeny ratio' (OR) for each genotype was quantified as follows: (minutes of daytime sleep in day 1) / (minutes of daytime sleep in day 4–5). Most wild-type flies, including our genetic controls, had a ratio in between 1.5 and 2.0 (*Figure 1B*). A lack of ontogenetic change would correspond to a ratio of 1.0, and a ratio below 1.0 indicates that young flies are sleeping less than mature flies. We thus set a ratio of 1.2 and below as our cutoff for defining hits.

## Circadian experiments

Flies were entrained to a 12:12 LD cycle for 3 days and then transferred to constant darkness (DD). Locomotor activity during days 1–7 in DD was analyzed with Clocklab software (Actimetrics, Wilmette, IL). Fast Fourier transform (FFT) values were calculated for all genotypes.

## Courtship assays

Virgin male flies were collected within 4 hr after eclosion and kept in isolation on regular food until being used in courtship experiments. Female Canton-S virgins (3–7 days post eclosion) were used in all courtship assays. A male and female were gently aspirated into a well-lit porcelain mating chamber (25 mm diameter and 10 mm depth) covered with a glass slide. Experiments were done in a temperature and humidity-controlled room at 25°C, 40–50% humidity. Courtship index (CI) was determined as the percentage of total amount of time a male was engaged in courtship activity during a period of 10 min or until successful copulation (*Siegel and Hall, 1979*). Courtship assays were recorded using a video camera (Sony HDR-CX405) and scored blind to experimental condition.

## TARGET system experiments

For the temporal mapping experiments using the TARGET system, flies were reared at 19°C (restrictive temperature) to prevent GAL80 denaturation, resulting in suppression of RNAi expression. For temporal windows during which *pdm3* knockdown was desired, flies were kept at 28°C (permissive temperature). Because of temperature-related changes in *Drosophila* developmental timing, flies were staged visually based on prior descriptions of developmental stages. Genetic controls were reared alongside experimental flies to control for temperature effects on development. All sleep experiments were conducted at 22°C in 12:12 LD cycles.

## Immunohistochemistry

Fly brains were dissected in 1X PBS, fixed in 4% PFA for 20 min at room temperature, and cleaned of remaining tissue in 1X PBS with 0.1% Triton-X 100 (PBS-T). Following 3 × 10 min washes in PBS-T, brains were incubated with primary antibody at 4°C overnight. Following 3 × 10 min washes in PBS-T, brains were incubated with secondary antibody for 2 hr at room temperature. After 3 × 10 min washes in PBST, brains were cleared in 50–70% glycerol and mounted in Vectashield.

## Imaging and analysis quantification

Brains were visualized with a TCS SP8 confocal microscope and images processed in NIH Fiji (*Schindelin et al., 2012*). All settings were kept constant between experimental conditions. Images were taken in 0.5 µM steps unless otherwise specified.

1. PER quantification

   To investigate PER expression in sLN$_v$s, sLNvs were labeled using an anti-PDF antibody and brains were co-stained with anti-PER. Flies were dissected at ZT0, 4, 6, 8, 12, 16 and 20. Area, mean gray value and integrated density of the PER signal was measured for each of the 4 sLNvs per hemisphere, defined by PDF staining. Corrected total cell fluorescence (CTCF) of the cell body was calculated using the formula: $CTCF_{Cell} = Integrated\ density_{Cell} - (Area_{Cell} \times Mean\ fluorescence_{Background})$ (*Dusik et al., 2014*).

2. TH+ innervation density

   Images were thresholded with the same settings across all images, and the thresholded area of TH+ neurites in the dFSB was measured. dFSB location was defined by use of anti-nc82 as

a general anatomical stain. Innervation density was calculated as (area occupied by TH+ neurites)/(area of dFSB).

3. 3D synapse counting (Brp-short$^{mCherry}$)

For each Z-slice, the FSB was selected and the surrounding signal was cleared. The full volume of the dFSB or vFSB was measured using the 3D Objects Counter function in Fiji with the settings: threshold = 1 min. puncta size = 1 Brp-short$^{mCherry}$ puncta were counted using the 3D Objects Counter function with the settings: threshold = 52 min.=2 max.=80.

4. CaLexA analysis

GFP fluorescence was quantified in a region of interest (ROI) based on single optical sections from whole-mount fly brains; mean GFP signal was normalized to fluorescent signal in an adjacent background ROI.

5. TH+ immunofluorescence

GFP fluorescence was quantified in a region of interest (ROI) based on single optical sections from whole-mount fly brains; mean GFP signal was normalized to anti-nc82 signal in the same ROI.

## RNA-Seq experiments

### Dissection/RNA extraction

40 brains per sample at the mid-pupal stage were dissected in cold AHL (108 mM NaCl, 5 mM KCl, 2 mM CaCl$_2$, 8.2 mM MgCl$_2$, 4 mM NaHCO$_3$, 1 mM NaH$_2$PO$_4$-H$_2$O, 5 mM trehalose, 10 mM sucrose, 5 mM HEPES). Four biological replicates (each with 40 brains) were dissected per genotype. Brains were transferred to 1 ml of Trizol and incubated for 5 min at room temperature (RT). 0.2 ml of chloroform was added and samples were inverted. Samples were incubated 2–3 min at RT, then centrifuged at 12000 g for 15 min at 4°C. Genomic DNA was removed using a gDNA eliminator column (RNeasy Plus Micro Kit, Qiagen). RNA was then extracted using the RNeasy MinElute Cleanup Kit (Qiagen).

### RNA library preparation and sequencing

Sequence libraries for each sample were synthesized using the NEBNext Ultra II Directional RNA kit following supplier recommendations and were sequenced on Illumina HiSeq-4000 sequencer as single-reads of 100 base reads following Illumina's instructions. The quality of the data was analyzed using fastqc v0.11.8 and multiqc v1.0.dev0.

### Differential gene expression analysis

The sequenced reads were mapped to the *Drosophila melanogaster* genome assembly dm6 using STAR v2.7.0f (**Dobin et al., 2013**). STAR was run with the default parameters with the following exceptions: —outFilterMultimapNmax 1 and —twopassMode Basic. The aligned reads were assigned to genes and counted using featureCounts (v1.6.4) run with default options on the dmel-all-r6.27 version of the *Drosophila melanogaster* annotation. Differential gene expression was performed on the gene count data in R v3.4.2 using DEseq2 v1.22.2 (**Love et al., 2014**). The annotated genes exhibiting an adjusted-P value <0.05 and |log2FC| > 1.0 were considered differentially expressed compared to control. Visualization of differentially expressed genes was done using the R-package ggplot2 v3.2.0.

### Gene set enrichment analysis

Gene set collections for Gene Ontology annotations were downloaded from public sources (http://www.go2msig.org/cgi-bin/prebuilt.cgi?taxid=7227). For comparison between the experimental and control groups, a gene signature was generated by ranking all expressed genes according to the DEseq2-derived test statistics. Enrichment analysis was performed with GSEA v3.0 (**Subramanian et al., 2005**) using weighted statistical analysis.

## CHiP-Seq analysis

We accessed the ENCODE database eGFP-pdm3 CHiP-Seq experiment (ID ENCSR518FUJ, contributed by K. White) and downloaded the ENCFF757XFT.bed file (optimal IDR thresholded peaks) (*Davis et al., 2018*; *ENCODE Project Consortium, 2012*). CHiP-Seq protocol and analysis workflow is detailed at the ENCODE website (https://www.encodeproject.org/experiments/ENCSR518FUJ/). Peaks were sorted by 1) p-value and 2) Signal value. Peaks were annotated using the HOMER (*Heinz et al., 2010*) annotatePeaks function with version dm6 of the *Drosophila melanogaster* genome.

## Generation of QUAS constructs

QUAS-WALIUM20 vector was obtained from J. Zirin at the Fly Transgenic RNAi Project (*Perkins et al., 2015*). The HMJ21205 pdm3 RNAi hairpin, originally used to generate the UAS-pdm3 RNAi construct (BSC #53887), was cloned into the QUAS-WALIUM20 vector using the pWALIUM20 cloning protocol (available at www.flyrnai.org). Briefly, the following oligonucleotides were synthesized and annealed (21 NT hairpin sequence shown in capital letters): 5' ctagcagtCAGCAACATTGTGAAGCGAGAtagttatattcaagcataTCTCGCTTCACAATGTTGCTGgcg 3' and 5'aattcgcCAGCAACATTGTGAAGCGAGAtatgcttgaatataactaTCTCGCTTCACAATGTTGCTGactg-3'. The QUAS-WALIUM20 vector was linearized by NheI and EcoRI, and the DNA fragment containing the hairpin was ligated into the vector. DNA injection was prepared with the Midiprep Kit (Qiagen). Injections were performed by Rainbow Transgenic Flies, Inc for production of transgenic flies at the attP40 and VK00033 landing sites.

## Statistical analysis

All analysis was done in GraphPad (Prism). Individual tests and significance are detailed in figure legends.

## Acknowledgements

We would like to thank members of the Kayser lab and members of the Penn Chronobiology and Sleep Institute for helpful discussions/input, and the Next Generation Sequencing Core (University of Pennsylvania) for sequencing/analysis support. *Figure 3A* was created with BioRender.com. We thank Salina Yuan for assistance with creating *Figure 8B*.

## Additional information

### Funding

| Funder | Grant reference number | Author |
| --- | --- | --- |
| National Institutes of Health | K08 NS090461 | Matthew S Kayser |
| National Institutes of Health | DP2 NS111996 | Matthew S Kayser |
| National Institutes of Health | T32 HL007953 | Leela Chakravarti Dilley |
| National Institutes of Health | F31 NS105447 | Leela Chakravarti Dilley |
| Burroughs Wellcome Fund | | Rajan Jain<br>Matthew S Kayser |
| March of Dimes Foundation | | Matthew S Kayser |
| Alfred P. Sloan Foundation | | Matthew S Kayser |

The funders had no role in study design, data collection and interpretation, or the decision to submit the work for publication.

### Author contributions

Leela Chakravarti Dilley, Conceptualization, Data curation, Formal analysis, Supervision, Funding acquisition, Validation, Investigation, Visualization, Methodology, Project administration; Milan Szuperak, Data curation, Formal analysis, Validation, Investigation, Methodology; Naihua N Gong,

Investigation; Charlette E Williams, Data curation, Investigation; Ricardo Linares Saldana, Data curation, Formal analysis; David S Garbe, Data curation, Investigation, Methodology; Mubarak Hussain Syed, Rajan Jain, Resources, Supervision; Matthew S Kayser, Conceptualization, Data curation, Formal analysis, Supervision, Funding acquisition, Investigation, Visualization, Methodology, Project administration

### Author ORCIDs

Leela Chakravarti Dilley (iD) https://orcid.org/0000-0001-8115-6821
Ricardo Linares Saldana (iD) http://orcid.org/0000-0003-2657-825X
Mubarak Hussain Syed (iD) http://orcid.org/0000-0003-2424-175X
Matthew S Kayser (iD) https://orcid.org/0000-0003-2359-4967

### Decision letter and Author response

Decision letter https://doi.org/10.7554/eLife.52676.sa1
Author response https://doi.org/10.7554/eLife.52676.sa2

## Additional files

### Supplementary files

• Supplementary file 1. Top gene expression changes (p-adj <0.05 and Fold Change > 1.2) from RNA-Seq analysis of mid-pupal brains in *Elav-GAL4 >pdm3 RNAi* and controls.

• Transparent reporting form

### Data availability

RNA Sequencing data is available at the NCBI GEO database (GSE147337).

The following dataset was generated:

| Author(s) | Year | Dataset title | Dataset URL | Database and Identifier |
|---|---|---|---|---|
| Chakravarti Dilley L, Saldana RL, Jain R, Kayser MS | 2020 | RNA-Seq of whole Drosophila brains (mid-pupal stage) with panneuronal pdm3 knockdown and controls | https://www.ncbi.nlm.nih.gov/geo/query/acc.cgi?acc=GSE147337 | NCBI Gene Expression Omnibus, GSE147337 |

The following previously published dataset was used:

| Author(s) | Year | Dataset title | Dataset URL | Database and Identifier |
|---|---|---|---|---|
| White K | 2016 | ENCODE Project Consortium | https://www.encodeproject.org/experiments/ENCSR518FUJ/ | ENCODE, ENCSR518FUJ |

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
