## [Decision Letter]

**Acceptance summary:**

It has long been appreciated that flies, like mammals, sleep more early in life. This paper represents a significant advance in our understanding of this behavioral plasticity for a number of reasons: the authors localize the window of *Pdm3* function, they find that it is critical for regulating connections between dopamine neurons and the sleep promoting fan-shaped body, and they identify *Msp300* as a potentially relevant target of *Pdm3*.

**Decision letter after peer review:**

Thank you for submitting your article "Identification of a molecular basis for the juvenile sleep state" for consideration by *eLife*. Your article has been reviewed by three peer reviewers, including Leslie C Griffith as the Reviewing Editor and Reviewer #1, and the evaluation has been overseen by Eve Marder as the Senior Editor. The following individual involved in review of your submission has agreed to reveal their identity: Alex C Keene (Reviewer #3).

The reviewers have discussed the reviews with one another and the Reviewing Editor has drafted this decision to help you prepare a revised submission.

Essential revisions:

1) The expression of data as ontogeny ratio seems to be masking complex phenomena. Ratios can change in the same direction via very different effects on the underlying sleep processes. All of the reviewers had similar concerns. The authors need to show all the raw data going into their ratio calculations. It is not at all clear for many of the manipulations that the baseline effects of things that move the ontogeny ratio in the same way also affect baseline sleep at different ages in the same way. The raw sleep data should also be analysed properly via appropriate multiple comparison tests, not just selected paired T tests.

2) The evidence that PDM3 acts via MSP300 is indirect without some more precise manipulation. This is a very interesting part of the story and enhancing the evidence of this claim could be done in many ways. The very compelling (but hard!) experiment would be removal of the PDM3 binding sites from the *msp300* gene to show it blocks the effects of PDM3. Another possible avenue would be to show more directly that MSP300 and PDM3 act in the same temporal and spatial window to exert their effects and have the expected epistatic relationship. The authors are encouraged to strengthen this part of the story.

---

## [Author Response]

Essential revisions:1) The expression of data as ontogeny ratio seems to be masking complex phenomena. Ratios can change in the same direction via very different effects on the underlying sleep processes. All of the reviewers had similar concerns. The authors need to show all the raw data going into their ratio calculations. It is not at all clear for many of the manipulations that the baseline effects of things that move the ontogeny ratio in the same way also affect baseline sleep at different ages in the same way. The raw sleep data should also be analysed properly via appropriate multiple comparison tests, not just selected paired T tests.

We agree with the reviewers, and now present the raw data as suggested (Figure 1—figure supplement 1, Figure 6—figure supplement 2, Figure 8—figure supplement 2). In all cases we present the data in graphical form to emphasize the raw sleep duration values, which speaks to the point raised. This analysis underscores (for example Figure 6—figure supplement 2A) that despite differences in sleep duration depending on control background, flies lose juvenile sleep with PDM3 knockdown.

Regarding statistical analysis, we apologize for any confusion. We did not use selected paired/individual t tests, but rather multiple t-tests with the Holm-Sidak correction for multiple comparisons.

2) The evidence that PDM3 acts via MSP300 is indirect without some more precise manipulation. This is a very interesting part of the story and enhancing the evidence of this claim could be done in many ways. The very compelling (but hard!) experiment would be removal of the PDM3 binding sites from the msp300 gene to show it blocks the effects of PDM3. Another possible avenue would be to show more directly that MSP300 and PDM3 act in the same temporal and spatial window to exert their effects and have the expected epistatic relationship. The authors are encouraged to strengthen this part of the story.

We agree with this point, and have now tested the spatial relationship directly using nervous system-wide knockdown of PDM3 (Elav QF2 > Q *pdm3* RNAi) along with specific knockdown of Msp300 in the central complex cells of interest (R93F07-GAL4 > UAS-Msp300 RNAi, UAS Dcr2). Our prediction was that this manipulation would suppress the *pdm3*-related sleep ontogeny phenotype. Interestingly, our results did not support this hypothesis, as sleep ontogeny remains impaired despite *Msp300* knockdown in these cells. This result suggests a more complicated interplay between PDM3 and *Msp300* with regard to sleep circuit wiring; for example, there might be a combination of cell-autonomous and non-cell-autonomous interactions. Alternatively, this result could reflect technical limitations of the approach (GAL4 expression pattern changes during development, poor *Msp300* knockdown efficacy with this driver, etc.). Regardless, this is an intriguing result that we will pursue in more detail in future work. We have added this result and discussion of it to the manuscript.